# Glycan-Lectin Interactions as Novel Immunosuppression Drivers in Glioblastoma

**DOI:** 10.3390/ijms23116312

**Published:** 2022-06-05

**Authors:** Angelica Pace, Fabio Scirocchi, Chiara Napoletano, Ilaria Grazia Zizzari, Luca D’Angelo, Antonio Santoro, Marianna Nuti, Hassan Rahimi, Aurelia Rughetti

**Affiliations:** 1Laboratory of Tumor Immunology and Cell Therapy, Department of Experimental Medicine, “Sapienza” University of Rome, Viale Regina Elena 324, 00161 Rome, Italy; angelica.pace@uniroma1.it (A.P.); fabio.scirocchi@uniroma1.it (F.S.); chiara.napoletano@uniroma1.it (C.N.); ilaria.zizzari@uniroma1.it (I.G.Z.); marianna.nuti@uniroma1.it (M.N.); hassan.rahimi@uniroma1.it (H.R.); 2Department of Neurology and Psychiatry, Neurosurgery, “Sapienza” University of Rome, Viale dell’ Università 30, 00185 Rome, Italy; lucadangelo_80@hotmail.com (L.D.); antonio.santoro@uniroma1.it (A.S.)

**Keywords:** glioblastoma, immunosuppression, C-type lectins, galectins, Siglecs, MGL/CLEC10A, galectin-9, extracellular vesicles (EVs), *O*-glycosylation, *N*-glycosylation

## Abstract

Despite diagnostic and therapeutic improvements, glioblastoma (GB) remains one of the most threatening brain tumor in adults, underlining the urgent need of new therapeutic targets. Lectins are glycan-binding proteins that regulate several biological processes through the recognition of specific sugar motifs. Lectins and their ligands are found on immune cells, endothelial cells and, also, tumor cells, pointing out a strong correlation among immunity, tumor microenvironment and vascularization. In GB, altered glycans and lectins contribute to tumor progression and immune evasion, shaping the tumor-immune landscape promoting immunosuppressive cell subsets, such as myeloid-derived suppressor cells (MDSCs) and M2-macrophages, and affecting immunoeffector populations, such as CD8^+^ T cells and dendritic cells (DCs). Here, we discuss the latest knowledge on the immune cells, immune related lectin receptors (C-type lectins, Siglecs, galectins) and changes in glycosylation that are involved in immunosuppressive mechanisms in GB, highlighting their interest as possible novel therapeutical targets.

## 1. Introduction

Gliomas are malignant brain tumors arising from transformed primitive neural stem cells, glial progenitor cells or dedifferentiated mature cell type [1]. Recently, the World Health Organization (WHO) released the newest 2021 classification of brain tumors, based on not only canonical criteria, such as histological and molecular features, but also novel diagnostic technologies, such as DNA methylome profiling. According to that, glioblastoma (GB) is characterized by the IDH-wildtype phenotype, while his mutant counterpart is identified as Astrocytoma IDH-mutant (grades 2–4) [2].

GB is the most common malignant brain tumor in adults, mostly in males compared to females. Nowadays, the first line of therapy requires maximal safe resection followed by radiotherapy (RT) and a high-dose of chemotherapy with temozolomide (TMZ). Despite the heavy treatment, most of the patients experience a recurrence, and the median overall survival remains below 18 months [3]. At recurrence, the standard of care is less well defined. Indeed, therapeutic options include afresh surgery and reirradiation in combination with systemic therapies, such as chemotherapy (TMZ, lomustine, carmustine) or anti-VEGF monoclonal antibody Bevacizumab [3,4,5]. Emerging therapies, such as immunotherapy and tumor-treating fields (TTFields), are available. TTFields and their ability to disrupt specifically mitotic process in cancer cells had shown promising results in prolonging the survival time, and several clinical trials are ongoing to study its safety and tolerability for further use [3,6]. On the other hand, immunotherapeutic approaches as immune-checkpoint inhibitors (ICIs) or personalized vaccination did not show in GB patients the same efficacy and benefits in terms of overall survival and improvement of prognosis as for other neoplasms [7,8,9,10]. Resistance to common therapies and failure of immunotherapeutic approaches are strongly associated to GB aggressiveness and microenvironment. Tumor cells within stromal cells cooperate in the establishment of a tumor-promoting microenvironment through the induction of immunosuppressive cytokines (i.e., TNF-α, interleukin (IL)-1α/β, IL-6), metalloproteases, several molecular pathways involved in angiogenesis, hypoxia and cell proliferation [11,12,13]. Moreover, several immunosuppressive cell subsets, such as tumor associated macrophages (TAMs) and myeloid-derived suppressor cells (MDSCs), are involved in supporting tumor evasion and resistance [14].

Glycosylation is a post-translational modification of proteins and lipids mostly expressed on the cellular membrane, ruling several physiological mechanisms such as cell metabolism, cell adhesion, cell–matrix interactions, cell–cell communication and immune homeostasis [15]. In tumors, aberrant glycosylation processes occur, leading to the generation of tumor-associated glycans as truncated and more intensely glycosylated structures for *O*-glycosylation or increased branching of *N*-glycans [16,17,18]. Lectins are a large family of proteins involved in the recognition of physiological and tumor associated-carbohydrates and they can be expressed by immune cells. Several studies have shown the role of glycan–lectins interactions in immunosuppressive mechanisms that occur in tumor immune escape [19,20,21,22,23].

In this review, we will discuss the state-of-the-art regarding immunosuppressive cell subsets, glycosylation changes and lectins as possible factors involved in immunosuppressive mechanisms and potential target for GB treatment.

## 2. Glioblastoma and Unique Immunosuppressive Networks

### 2.1. The GB Microenvironment

Tumor development is related not only to cancer cells, but also to the interactions with surrounding cells and the microenvironment. Indeed, tumor microenvironment (TME) actively sustains tumors, and at the same time, is directly or indirectly modulated by the tumor itself. It contributes to the stabilization of immune and drug resistance mechanisms [24,25]. Although GB is highly vascularized, the circulation is inefficient, triggering the generation of a hypoxic microenvironment [26]. Hypoxic regions are disseminated in the entire tumor and different oxygen levels can be associated to distinct cell types [27]. It has been shown that hypoxia-induced factor 1 (HIF-1α) promotes the expansion of the CD133^+^ glioma stem cells (GSC), which cooperate in the tumor resistance to therapies [28]. Moreover, HIF-1α regulates the transcription of several genes involved in anaerobic and aerobic glycolysis, lipid metabolism, autophagy and angiogenesis [29]. Hypoxia promotes the adaptation of cancer cells through a metabolic reprogramming of aerobic glycolysis and uptake of glucose. This results in the protection of tumor cells from cell damage, in the abnormal proliferation and in the development of resistance, facilitating malignant progression [29]. In addition, HIF-1α sustains cell migration facilitating epithelial-mesenchymal transition (EMT) through Snail proteins and ECM remodeling and increasing tumor invasive potential by dysregulating glycosylation processes and altering the blood vessel permeability [29,30]. Hypoxic environment is firmly linked with angiogenesis, where HIF-1α stimulate the expression of pro-angiogenic genes, such as vascular endothelial growth factor (VEGF), placenta-like growth factor (PlGF), platelet-derived growth factor (PDGF), angiopoietin (Ang)-1 and -2; erythropoietin (EPO) and insulin-like growth factor 2 (IGF2) [27,29,31]. VEGF acts as a key regulator of angiogenesis modulating blood vessel formation and permeability and providing mitogenic and survival stimuli for endothelial cells [32]. VEGF also contributes to the stabilization of immunosuppressive networks that are a typical feature of GB immune microenvironments [30,33]. Lastly, hypoxia has multiple direct effects on the immune system, triggering several mechanisms besides VEGF production, such as immune suppressive myeloid cell shift and recruitment, impairing T cell response, thus promoting a complex immunosuppressive network, fostering tumor progression and poor prognosis for patients [29,31,33].

### 2.2. Immune Cells in GB Microenvironment

Notoriously, GB has been considered a “cold” tumor due to its privileged location and the presence of few immunoregulatory cells, mostly immunosuppressive actors [34]. Actually, the “privileged term” has been reconsidered after the discovery of lymphatic vessels outlining the dural sinuses in mice and potentially analogous structures can exist in the human brain, underlining how it is possible for the immune system to enter and interact with the central nervous system (CNS) [35]. Moreover, the blood–brain-barrier (BBB), which naturally separates the brain from the systemic circulation, is an obstacle for the delivery of therapeutic agents. In a tumor context, BBB can lose its integrity due to microvascular sprouting, new vasculature branching and dysfunction of tight junctions, allowing the infiltration of inflammatory cells [36]. Several studies have shown the presence of different immune subsets with distinct localization and function in GB microenvironments [37,38]. Within the TME composition, GB cells release a variety of chemoattractive factors such as CCL2, CXCL12, and SDF-1 that actively promote the recruitment of immune cells [39].

Myeloid cells are the most represented cell type in GB microenvironment (Figure 1) [14,40].

Among them, TAMs and MDSCs are the most diffused myeloid cells that cooperate in immunosuppressive mechanisms and tumor resistance. TAM population includes infiltrating bone marrow-derived macrophages (BMDMs) and microglia (MG), that make up to 30–50% of the total tumor mass [39,41]. MG are brain resident and resting macrophages that are involved in CNS development and homeostasis and immune surveillance [42]. MG and BMDMs are distinguished by the expression of surface markers, as CD49d and CD45: MG is usually identified as CD45^low^ CD11b^+^ CX3CR1^+^ P2RY12^+^ CD49d^−^ while macrophages are CD45^high^ CD11b^+^ CX3CR1^+^ CD49d^+^ P2RY12^−^ [43]. Recently, the transmembrane protein 119 (TMEM119) has been reported as novel unique marker of resident MG both in humans and mice [44]. Moreover, MG and BMDMs differs for their localization underlining their distinct contribution to immunosuppression. MG cells appears stationary, large and branched and infiltrate the white matter, outlining the edge of the tumor, while BMDMs are usually localized near blood vessels in central and necrotic area of the tumor [14,43]. MG takes part in the tumorigenesis process through TGF-β release, enhancing the immunosuppression in the TME by blocking T-cells and NK cells activity and promoting T regulatory cells (Tregs) [38,45]. In response to chemokines and cytokines released by GB cells, BMDMs are recruited from the periphery to the TME. By interacting with tumor cells and TME, BMDMs change their expression profiles and mostly acquire a M2 pro-tumor phenotype through a modulation of immunosuppressive and phagocytic mechanisms and oxidative and iron metabolisms [14,46]. A recent study on patients-derived GB sections revealed the high infiltration of CD163^+^ cells (M2-like TAMs) in both the tumor core (TC) and peritumoral area (PTA). Interestingly the high frequency of these cells in the TC correlate with a significant prevalence of immunosuppressive markers, such as IDO and PD-L1, underlining a pro-tumoral immune system within GB TME [47]. Moreover, in a recent study the selective targeting of TAMs by inhibiting CSF-1R resulted in the alteration of macrophage polarization and block of glioma progression, proposing TAMs as potential therapeutic targets [48].

MDSCs play a fundamental role in promoting tumor progression impairing T-cell functions and inducing Treg cells [49]. MDSCs consist of two large group of cells, i.e., monocytic MDSCs (M-MDSCs) and polymorphonuclear MDSCs (PMN-MDSCs), which are differentiated by molecular and phenotypical features [14].

In the tumor site, MDSCs functions are strongly controlled by several soluble factors, such as IL-6, IL-10, TGF-β, CCR2, VEGF secreted by GB cells and immune cells, leading to the modulation of molecular pathways mediated by Janus tyrosine kinase (JAK) and signal transducer and activator of transcription 1, 3 and 6 (STAT1; STAT3; STAT6). This results in the promotion of MDSCs proliferation and survival and induction of immunosuppressive molecules, such as arginase 1 (ARG1), reactive oxygen species (ROS) and inducible nitric oxide synthase (iNOS). Thus, high levels of circulating ARG1 in GB patients’ blood have been considered as markers of MDSCs activation [50].

Several studies have reported the contribution of both MDSCs subpopulations to the progression of gliomas and high levels of circulating MDSCs are associated with the worst prognosis for GB patients [50,51,52,53,54]. Indeed, PMN-MDSCs have been identified as the main circulating MDSC subpopulation in GB patients’ blood and their proportion is higher than M-MDSC in GB tumors [55,56,57]. Moreover, Gielen et al., showed that MDSC levels of GB patients’ blood were significantly higher than healthy controls and found that PMN-MDSCs were representative of MDSC infiltrating tumor mass [58]. The discovery of novel markers, such as LOX1, helps to further distinguish MDSC subpopulations and elucidate their role and mechanisms in tumor progression [59]. Indeed, a recent study showed that LOX1^+^ identifies a specific PMN-MDSC subpopulation that is increased in GB patients’ blood and tumor tissue and is characterized by a high expression of ARG1 and iNOS and strong immunosuppressive ability to inhibit CD3^+^ T cells [52].

## 3. GB and Glycosylation Pathways

### 3.1. Glycosylation Pathways

Glycosylation is the main post-translational modification of proteins and lipids, that requires a highly monitored enzymatic complex network from nucleus to Golgi apparatus [60,61]. Indeed, specific enzymes participate to the finely tuned phases of the glycosylation process, i.e., core extension, elongation and branching, and capping of the carbohydrate chains. This generates an astonishing diversity in the glycome that corresponds to highly specific and selective biological functions [60,61].

Most of the glycoconjugates and glycans are exposed on the cellular membrane and play a crucial role in tissue lubrification and protection, cell–matrix interactions and cell signaling [62]. Glycans are also involved in immunomodulation mechanisms through the interaction with glycan-binding proteins of lectins family, regulating immune homeostasis and host–pathogen interactions [63].

*N*- and *O*-linked glycans are the most common post-translational modification (PTMs) in extracellular components (Figure 2).

*O*-linked glycosylation is characterized by the addition of monosaccharides, as such as *N*-acetyl-D-galactosamine (GalNAc), D-glucose (Glc) or D-mannose (Man), to serin or threonin residues (Ser, Thr). *O*-linked Glc, Fucose (Fuc) or Man are found on the EGF-related *O*-glycans. *O*-linked GalNAc is the substrate for the extension of long polylactosamine side chains in the so called “Mucin-type glycosylation”. In this pathway, the synthesis of GalNAc α1-*O*-Ser/Thr (the Tn antigen) is required for the next synthesis of the T antigen (Galβ1-3GalNAcα1-*O*-Ser/Thr) and further extension of long polylactosamine side chains that decor the protein backbone of mucins. These molecular structures function as protection and lubrification of epithelia and become truncated in carcinoma. Additionally, *O*-linked xylose may trigger the elongation of glycosaminoglycans (GAGs) on the backbone of proteoglycans. In response to nutrient and stress response, addition of *O*-linked *N*-Acetyl-D-glucosamine (*O*-GlcNAcylation) can occur on many intracellular proteins [64].

*N*-linked glycosylation consists in the covalent binding of the oligosaccharide *N*-Acetyl-D-glucosamine (GlcNAc) to the asparagine (Asn) residue of polypeptide chains mediated by the oligosaccharyltransferase (OST) complex. Elongation processes give rise to a common pentasaccharide core region composed by GlcNAc and Man residues. This *N*-core structure is the substrate for the branching of the side chains that are further diversified in carbohydrate composition. On the basis of that, *N*-glycoconjugates are classified in high mannose, hybrid or complex [65,66]. High mannose structures are composed only by Man, while complex structures are made also by other monosaccarides (GalNAc, Fuc, Galactose, Sialic Acid (Sia)). Hybrid structures are characterized by the presence of both high mannose and complex structures on distinct branches [65,66].

Lipid glycosylation is also crucial for the biology of cells. In particular, glycosphingolipids (GLS), which are the major component of the outer cell plasmamembrane, are composed by a hydrophobic ceramide backbone linked to a first glycan moiety (Gal or Glc) through a glycosid linkage [67]. A complex enzyme network located among the intracellular compartments mediates the subsequent addition of other carbohydrate residues as such as GlcNAc, GalNAc, Fuc and Sia generating structurally different molecules, which potentially correspond to a distinct function. Gangliosides (GG) are acidic GLS containing sialic acid residues, enriched in cell membrane microdomains, and are more abundantly found in the nervous system [68,69].

Despite the diversity of glycosylation pathways, the carbohydrate chains can undergo to non-specific modifications thus generating molecular structures shared by distinct glycoconjugates. Sulfation, the addition of SO_3_ group to the carbohydrate backbone, is the most abundant glycan modification generating a large pattern of sulfated structures that play wide essential biological role. Moreover, the terminal addition of Sia (Sialylation) or Fuc (Fucosylation) are crucial events in the capping of the glycan chains, thus blocking the elongation process and generating glycan structures specifically recognized by glycan-binding proteins [60,70].

The variety and complexity of the glycome is shaped by the cellular stress and becomes a strategy for the cell to modulate its function and interactions with the microenvironment. Indeed, glycosylation alterations may be regarded as key features of diseases and a key factor for immune recognition [71].

### 3.2. Aberrant Glycosylation Processes in GB

Aberrant changes in the glycosylation machinery early occur during tumor transformation, i.e., increased *N*-glycan branching, truncated *O*-glycans and augmented *O*-glycan density on the protein backbone, and synthesis of neo-glycans carrying numerous terminal Sia or Fuc [19,72]. These aberrant glycosylation processes generate tumor associated glycans that play a key-role in tumorigenesis, metastasis and immune modulation [72,73].

Thus far, an accumulating body of evidences indicate that glycosylation changes occur in GB and involve a large variety of cellular substrates (proteins and lipids) and distinct metabolic pathways, thus contributing to the complex biology and impaired immunogenicity of GB (Figure 3) [18,74].

In GB, changes in *N*-glycosylation are characterized by altered branching, increased fucosylation and sialylation. Several receptors may be aberrantly *N*-glycosylated and this impacts on cell functions. *N*-acetylgalactosyltransferases (GnTs) are key enzymes for the *N*-glycan synthesis. MGAT1 (GnT1) and MGAT5 (GnT5) are up-regulated in glioma cells and in GB tissues [75,76]. MGAT1 plays a crucial role in the conversion of the carbohydrate core from High mannose to complex/hybrid structures.

This change increases the stability of the GLUT1 transporter, which is *N*-glycosylated, promoting cell proliferation [75]. MGAT5 catalyzes the addition of terminal β-1,6-GlcNAc to the *N*-glycan and its overexpression in cancers supports *N*-glycan branching and correlates with tumor invasion [76,77]. Increased *N*-branching augments terminal sialylation and fucosylation processes that contributes to tumor spreading [78].

The tumor associated Sialyl-Lewis Acid (S-Lewis X) glycan generated by these changes is recognized by immune lectins [79]. Dysregulated fucosylation is a feature associated to GB aggressiveness. Fucosylation of the first GlcNAc residue of the *N*-core structure (Figure 3) is catalyzed by the fucosyltrasferases 8 (FUT8). In human GB FUT8, overexpression is associated with high tumor grade and aggressive disease. Interestingly, MET and EGFR molecules are substrate for FUT8 and the altered glycosylation increases their function with an overall gain of motility and proliferation for the tumor cells [80].

Indeed, inhibition of *N*-linked glycosylation has been proposed as an appealing pharmacological strategy to dampen receptor kinase activity and enhance radiosensitivity of GB cells [81].

The increased sialylation levels observed in glioma cells are mainly due to the overexpression of the α-2,3-sialyl transferase (ST3GAL1). Its up-regulation correlates with poor prognosis in GB patients and is associated to enhanced invasive potential [82]. On the contrary, α-2,6-sialylation is downregulated in the tumor cells, but it is upregulated in the GB vasculature, suggesting its crucial role for endothelial survival [83,84,85]. Therefore, two distinct sialylation pathways underlie distinct biological mechanisms for GB progression [74]. Little is known about changes in the *O*-linked mucin type glycosylation. In carcinoma, alterations of this enzymatic pathway generate truncated *O-* glycans, i.e., Tn and T and their sialylated version STn (NeuAcα2-6GalNAcα1-*O*-Ser/Thr) and ST (NeuAcα2-3Galβ1-3GalNAcα1-*O*-Ser/Thr) [86]. In GB, alteration of the enzymatic *O*-glycosylation pathway was found and Tn glycan expression was revealed by the binding of anti-Tn lectins in mouse model as well as in human tumor cells [87,88]. In GB mouse model, the overexpression of Tn has been associated to strong immunosuppression (as described in Section 4.1) [88].

Additionally, glycolipid components are altered in GB. Early work had shown changes in GLS components in human gliomas, in particular ganglioside GD3 was shown to be increased in gliomas and correlated with malignancy [89,90]. Thus far, increase of GD2 and GD3 in GB has been associated with resistance to therapy and increase on invasiveness [91,92]. The aberrant GG pattern alters the organization of membrane microdomains with huge impact on the activity of tyrosin-kinase receptors and downstream cell signaling [93]. Indeed, GD3 plays a crucial role in the stemness and tumorigenicity of GB by activating c-MET signaling [94]. At the same time, GD3 has been shown to modulate innate immune response by specifically interacting with Siglec7 (see Section 4.2). More recently, increase of *O*-Ac-GD3 (GD3 carrying acetylated Sia) and the ganglioside GT1 were found in human GB samples, while GM1 was found in peritumoral brain tissue [95,96]. These differential patterns may be relevant to both tumor biology and immunomodulation mediated by GGs [97].

Another crucial source of glycan diversity is the matrisome. In GB, the ECM is increased due to the up-regulated secretion of PGs and the associated *O*-linked GAGs [98,99,100]. These ECM components can undergo to aberrant sulfation processes that appear to be relevant for the glycan alteration in GB ECM. Modulation of sulfation among distinct PGs was found and several core matrisome PGs (as such as decorin, agrin, glypcan-1), and laminin, tenascin, fibronectin among the others, were differentially regulated in tumor vs. control tissues. Interestingly, changes in PG composition were also found within the GB samples in accordance to the IDH mutant status. From this analysis, *O*-glycosylated peptides from PGs were found only in the GB tissue and not in the control, again indicating that dysregulated glycosylation process generates specific glyco-profile [101].

Indeed, a glycoproteomic fingerprinting would be of great interest to in depth characterize the ECM components to unveil biological mechanisms aiding cell growth and metastatic spreading that could be useful as potential therapeutic targets [74,102].

## 4. Glycan-Lectin Interactions and Immunosuppressive Networks in GB

The de novo exposed tumor glycan array triggers novel immune interactions mediated by lectins that are able to skew the immune system function with high impact on the overall tumor progression [103]. Lectins constitute a large protein family that share a common carbohydrate recognition domain (CRD) for the binding of specific glycosylated structures [19].

Human lectins are classified by considering their subcellular location and their structures. Lectins can be incorporate in the cellular membrane or have a soluble form and based on their form they can be divided into groups as C-type lectins, I-type lectins (including Siglecs), S-type lectins (also known as galectins), P-type lectins (known as Selectins) and pentraxins [104]. In tumor, they have been proposed to decipher the tumor glycocode that modulate immune tolerance/suppression [103].

In GB, the expression of lectins from the C-type, Siglec and galectin family has been described and their possible role in the stabilization of immune networks has been proposed [105,106,107]. The main glycan–lectin interactions found in GB are represented in Figure 4.

### 4.1. C-Type Lectins in GB

C-type lectins (CTLs) are a large family of proteins that share one or more conserved C-type lectin-like domain (CTLD) able to recognize a broad range of carbohydrate motifs. CTLs are mainly expressed on myeloid cells, and their expression is carefully regulated or induced under specific conditions, as inflammation. CTLs are mainly known for their role in antimicrobial immunity and tissue homeostasis, but they can both promote or suppress the immune response in disease context, such as cancer [23,108,109]. 

For example, alteration of MHC class I molecules due to oncogenic transformation or infection can be sensed by natural killer (NK) cells and can activate the C-type lectin NKG2D-mediated cytotoxicity. However, several C-type lectins, such as DC-SIGN, CLEC14A, macrophage galactose C-type lectin (MGL) contribute to tumor progression inducing immunosuppressive responses by sensing abnormal or altered tumor-associated carbohydrates [110]. Thus far, the few data on the expression and role of CTLs in brain diseases suggest they may play an immunosuppressive role in the immunological networks. CTLs expression associated to microglia has been observed in neuroinflammatory degenerative diseases as Alzheimer disease (AD) and Multiple Sclerosis [111,112].

In these pathological settings, expression of CTL by microglia has been observed to mitigate the inflammatory reaction suggesting a possible protective role in inflammatory mediated neurodegenerative diseases. Expression of MGL by macrophages and microglia in experimental autoimmune encephalomyelitis (EAE) mouse model, induced apoptosis of autoreactive T cells and release of immunosuppressive IL-10, thus proposing a role of MGL as negative regulator of autoimmune-driven neuroinflammation [112]. MGL is a Type II C-type lectin, with specificity for the truncated Tn *O*-glycan (GalNAc-α-Ser/Thr) (Table 1) [113]. 

MGL engagement by its ligand triggers phenotypic and metabolic changes in the antigen presenting cells [114] that are finely tuned by the structure of the Tn carrying ligand [113]. In physiological context, MGL expressing APCs appears to act as negative regulator for naïve T cell homeostasis and Treg cells through the interaction with the Tn-glycosylated CD45RA [115]. Expression of MGL and its Tn *O*-glycan ligand has been described in GB human tissues, preferentially associated to tumor infiltrating CD163^+^ cells [88]. Results from a GB syngenic mouse model highly expressing the Tn *O*-glycan showed that distinct myeloid infiltrating cell subsets heterogeneously expressed MGL. Interestingly, in the MGL^+^ tissue areas, a strong infiltration of PD-L1^+^ TAMs was found [88]. These results support the hypothesis that the MGL-Tn axis may contribute to the immunosuppressive network in GB. It is interesting to recall that the Tn-MGL axis has been validated as a therapeutic target in an ovarian cancer mouse model by means of glycomimetic peptides [116,117].

**Table 1 ijms-23-06312-t001:** Classification, expression, binding preference, glycosylated ligand, known molecular mechanisms and role of C-type lectins and Siglecs in GB. Abbreviations: GalNAc: N-Acetyl-d-galactosamine; MUC1: mucin-1; MUC5: mucin-5; MUC16: mucin-16; MUC24: mucin-24; ERP44: Endoplasmatic Reticulum Resident Protein 44; LAMP1: Lysosome-associated membrane glycoprotein 1; LAMP2: Lysosome-associated membrane glycoprotein 2; QSOX1: Sulphydryl oxidase 1; SEL1L: Protein Sel-1 homolog 1; LRR8CD: Leucin-rich repeated-containing protein 8D; AGRN: Agrin; APP: Amyloid beta A4 protein; DAG1: Dystroglycan; FN1: Fibronectin; NID-2: Nidogen-2; PODXL: Podocalyxin; SDC3: syndecan-3; VCAN: Versican core protein; GBS β-protein: group B Streptococcus (GBS) β-protein; GD3: ganglioside.

Lectin	Expression	Recognized Carbohydrate Motif	Glycosylated Ligand	Molecular Mechanism	Role in GB	Ref.
**MGL**	DC and cDC2 Macrophages CD163^+^ cells Activated MG	GalNAc-α-Ser/Thr	CD45RA *Matrix/cell adhesion* (VCAN, SDC3, PODXL NID-2, FN1, DAG1, APP, AGRN) *Cell metabolism* (ERP44,LAMP1/2, QSOX1, SEL1L, LRR8CD) *Mucins* (MUC1/16/24)	Promotes ERK phosphorylation and NfkB Enhanced secretion of IL10 Activation TLR signaling Promotes DC	TAM/CD163^+^ cells mediated immunosuppression	[111,112,113,118,119]
**Siglec5**	Monocytes DC Neutrophils Macrophages	α-(2-3)-Sialic acid α-(2-6)-Sialic acid α-(2-8)-Sialic acid	GBS β-protein	ECM remodelling	MDSCs mediated immunosuppression	[63,73,104,107,120,121]
**Siglec7**	NK cells Monocytes Macrophages Mast cells DC	α-(2-3)-Sialic acid α-(2-6)-Sialic acid α-(2-8)-Sialic acid disialogangliosides	CD43 GD3	ECM remodelling	MDSCs mediated immunosuppression	[63,73,104,107,120,121]
**Siglec9**	NK cells DC T cells Neutrophils Macrophages Monocytes	α-(2-3)-Sialic acid α-(2-6)-Sialic acid α-(2-8)-Sialic acid	Glycophorin Hyaluronic acid MUC1/5	Modulation of MAPK/ERK Neutrophils inhibition/death M2 polarization ECM remodelling Inhibits macrophages phagocytosis	MDSCs mediated immunosuppression	[63,73,104,107,120,121,122,123]

### 4.2. Siglecs in GB

Siglecs belong to the I-type lectins immunoglobulin superfamily that mainly recognize sialic acids (Sia) and other carbohydrate residues. They are classified based on their structure in CD33-related Siglecs (Siglec3 or CD33; Siglecs 5-11; Siglecs14 and 16) and other Siglecs (Siglec1 or CD169; Siglec2 or CD22; Siglec4a or MAG and Siglec15).

Siglecs function as immunomodulatory receptors that mainly mediate immunosuppressive responses through the phosphorylation of the immunoreceptor tyrosine-based inhibition motif (ITIM) depending on the cell which they are expressed by and on the ligand they interact with. All Siglecs are expressed on immune cells, except for the myelin-associated glycoprotein (MAG) [104]. Siglec1 and 12 expression is restricted on macrophages, CD22 on B lymphocytes, Siglec16 on MG while other Siglecs, such as CD33 and Siglec7, can be found on several immune cell subsets [121,122,123,124].

MAG is selectively expressed in the nervous system and is known to have an important role in the maintenance of myelinated axons, the physiology of oligodendrocytes and the suppression of axonal regeneration after injury [125]. MAG exerts its function through the interaction with Nogo receptor and gangliosides containing 2,3-linked sialic acid [125]. Interestingly, in U87MG glioma cells, it has been shown that MAG is able to reduce the migratory capacity of these cells in culture [126].

It has been shown that in tumor models, sialoglycan–Siglecs interactions are able to impair and suppress the effector immune cells, such as NK and CD8^+^ T cells. Moreover, Siglec7 and 9 glycosylated-ligands have been found in several cancers, including GB [127]. In a recent work, the authors characterized the expression of Siglec5, 7, 9 on MDSC both from peripheral blood and glioma infiltrating cells and found the respective ligands both on GB cell lines and patient-derived glioma tissue samples (Table 1). These evidences underline the possible interaction between glioma cells and MDSC via the sialic acid–siglec axis, modulating their function and shaping the immunosuppressive TME [107].

Interestingly, Dusoswa et al. showed a high presence of *N*-linked and *O*-linked α-2,3 sialic acids and *N*-linked α-2,6 sialic acids on GB-derived extracellular vesicles (EVs). GB-derived EVs express Sia-glycoconjugates which are ligands for Siglec9, whereas no ligand for the other Siglecs was detected [128]. Siglec9 is mainly found on immune cells, as NK, T and DC and can mediate immunosuppressive functions through the activation of the two ITIM sequences comprise in the intracellular domain. In this work, the authors showed that the modification of EVs-glycosylation profile through enzymatic desialylation enhanced EV’s internalization by DCs, thus suggesting that glycosylation degree may indirectly modulate antigen processing, as shown for other glycan carrying EV systems [129]. Although, these results sustain the relevance of that Siglecs as sensors of sialylated carbohydrates in GB, further studies are required to elucidate their functional contribution to the immunosuppressive networks.

### 4.3. Galectins in GB

S-type lectins, or galectins, are defined by the presence of at least one conserved carbohydrate recognition domain (CRD) with high affinity for lactose or *N*-acetyllactosamine (LacNAc; Galβ1-4GlcNAc)-containing ligands. Currently, galectins are structurally differentiated in three groups, as they can exist as dimeric form (galectin-1, -2, -5, -7, -10, -11, -13, -14, -15), tandem-repeat form (galectin-4, -6, -8, -9) and the monomer or multivalent chimera type (galectin-3) [130].

Galectins can be retained in the cytosol or secreted both by transformed and normal cells, such as fibroblasts, endothelial and immune cells [131,132]. Galectins are involved in a wide range of basic cellular functions, as apoptosis, proliferation, cell adhesion and signaling, but also in immune system modulation and cancer development [130]. Interestingly, galectins exert their function both intracellularly and extracellularly [133].

Galectins have been shown to play a key role in tumor progression, promoting angiogenesis and ECM remodeling through the binding with highly glycosylated ligands expressed by several cell types. Tumoral galectins, interacting with glycosylated immune-related ligands, induce immunosuppression promoting tolerogenic DCs, CD4^+^ and CD8^+^ T cell apoptosis and exhaustion, FoxP3^+^ Treg cells expansion and NK impairment [21]. Upon interaction with galectins, glycoproteins and glycolipids undergo structural rearrangement, thus modulating their function and the downstream signaling pathways [133]. Galectin-1 shapes the immune compartment, promoting the proliferation of tolerogenic DCs and M2-macrophages, the apoptosis of Th1 and Th17 T cells and the expansion of Treg cells. Moreover, galectin-3 exerts its immunomodulatory functions affecting effector T cells activity by altering the immunological synapses organization and distancing TCR from CD8. These molecular events induce anergy, exhaustion and suppression of T cells. At the same time, galectin-3 promotes immunosuppressive cells, as Tregs and MDSCs, thus dampening the anti-tumor immune response [133,134,135].

The abundant galectins levels both in TME and peripheral blood have been proposed as prognostic factors and associated to a negative prognosis for oncologic patients [130,136]. Several galectins have been found in both neurons and glial cells and physiologically contribute to CNS development and functions. Alterations in galectins levels are observed in neuroinflammatory and neurodegenerative disease, and also in CNS tumors [105,137]. Indeed galectin-1, -3, -8 and -9 are expressed in GB and the interaction with their respective glycan-carrying molecules blocks the anti-tumor response, thus promoting immunosuppression (Table 2) [138].

Galectin-1 and -9 are the two galectins studied for their association with immunosuppressive mechanisms [21]. In vitro and mice model studies have shown that galectin-1 promotes tumor cells’ migratory capability and invasiveness through modification of cytoskeleton and small GTPases [138,139,140,141,142]. The selective silencing of glioma-derived galectin-1 delays tumor progression and prolongs the survival of glioma-bearing mice impairing angiogenesis and the recruitment of myeloid cells as MDSCs and macrophages [143]. Gou et al., showed that the new putative oncogene FAM92A1-289 is involved in DNA methylation in GB cell line and interacts with galectin-1 contributing to tumor progression [144]. Recently, LGALS1 gene, coding for the galectin-1, has been included among the genes that define a glioma microenvironmental gene signature and has been identified as a key immunosuppressive gene in GB [145]. 

Galectin-1 levels directly correlated with the grade of GB samples and its selective knockdown impaired immunosuppressive mechanisms by down regulating M2 macrophages and MDSC cells and by inhibiting immunosuppressive cytokines [145]. Expression of galectin-1 was also associated with a clinical response: GB patients treated with radiotherapy with a low expression of galectin-1 showed a better median survival than patients with high levels [146]. The silencing or knockdown of galectin-1 in vivo via intranasal administration or through siRNA-loaded chitosan lipid nanocapsule showed promising results by altering the GB TME and immune compartment and reducing TMZ resistance, unraveling galectin-1 importance in immunosuppression and tumor promotion [147,148,149].

Galectin-9 is known as eosinophil chemoattractant and, more recently, as a negative modulator of adaptative immune response. The major mechanism through which galectin-9 exerts its immunosuppressive mechanisms involves the binding to TIM3, that is widely expressed on immune cells, especially on T-cells [21]. Moreover, other immune checkpoints are recognized by galectin-9, therefore galectin-9 is extremely relevant for immunosuppressive mechanisms (see Table 2) [21]. Through a comprehensive analysis of 1027 glioma patients, Yuan et al., found a strong common upregulation of galectin-9 in GB compared to normal brain tissue. In addition, high galectin-9 expression levels correlate with a shorter overall survival in lower-grade glioma patients. In GB tissue samples, galectin-9 expression is also associated to immunosuppressive M2-macrophages and its levels positively correlate with immune checkpoint molecules [150,151]. Notably, galectin-9 also exerts its functions when release in EVs. A fascinating study underlined that GB derived-exosomes (GB-Exos), enriched in galectin-9, anontribute to tumor progression by impairing DCs and CD8^+^ T cells function by TIM3 binding. Indeed, exosomal galectin-9 activity on DCs is TIM3 dependent and knockout of TIM3 in DCs, restore DCs function and activation [152]. Taken together, these studies suggest that galectin-9/TIM3 can be one of the key mechanisms that sustains immunosuppression in GB TME and it has been proposed as therapeutical intervention for targeting cell–cell interaction and exosomes communication [153,154,155].

Galectin-3 and -8 role in GB progression is less well defined. Little is known about their immune-related functions, while a better knowledge on their interplay with ECM is available (see Table 2) [130,137]. Depending on its expression, galectin-3 is able to reinforce or weak cell adhesion and cell–matrix interactions promoting the activation of pathway involved in ECM remodeling and EMT [156,157]. Moreover, galectin-3 upregulation under hypoxic and nutrient deprivation has been correlated to tumor aggressiveness and poor prognosis in several cancer type [21,158]. Galectin-3 has been also proposed as biomarker for differential grading and diagnosis of gliomas [159], resistance to standard therapies and high levels are associated with a poor survival [160,161,162].

Lastly, galectin-8 contribution to GB is very poorly understood. It is interesting to note that galectin-8 is the only one of this family that shows an high affinity for α-2-3-sialylated glycans, of which GB is rich [85,163]. In a recent study was showed that galectin-8 promotes proliferation and prevents apoptosis of GB cell lines [164]. Indeed, galectin-8 is strongly implicated in cell adhesion, growth and cytoskeletal organization, thus underlining a possible favorable function in tumor development and progression [165]. Moreover, high levels of galectin-8 had been found in oncologic patients and had been often considered as a poor prognostic factor in several malignancy, as multiple myeloma and prostate cancer [166,167,168]. Interestingly the selective knockdown of galectin-8 in a mouse model showed a notable reduction of the tumor size and a minor metastatic process, highlighting its importance as potential therapeutical target. However, high levels of galectin-8 have been also associated to a better survival for gastric cancer patients, pointing out the need to better clarify galectins role in carcinogenesis [169].

**Table 2 ijms-23-06312-t002:** Classification, expression, binding preference, glycosylated ligand, known molecular mechanisms and role of galectins in GB. Abbreviations: N-acetyllactosamine: LacNAc; Galβ1-4GlcNAc; Forssman pentasaccharide: GalNAc-α-(1,3)-GalNAc-β-(1,3)-Gal-α-(1,4)-Gal-β-(1,4)-Glc; MCAM: Melanoma Cell Adhesion Molecule; LFA-1: Lymphocyte function-associated antigen 1; TLR-4: Toll Like Receptor 4; LAG-3: Lymphocyte activation gene 3 protein; VEGF-R2: Vascular endothelial growth factor receptor 2; Glut-2: Glucose transporter 2; tDCs: tolerogenic dendritic cells; c-Maf/aryl: transcription factor Maf/aryl hydrocarbon receptor; STAT/JAK1-2: Janus kinases 1-2 and activator of transcription proteins; GSK-3β: Glycogen synthase kinase 3 beta; NRP-1: neuropilin-1; IL-2: Interleuchin 2 receptor; TGF-β1: Transforming growth factor 1; TGF-βR: Transforming growht factor beta receptor; Lck: non-receptor tyrosine-protein kinase; Bat3: Large proline-rich protein BAT3.

Lectin	Expression	Recognized Carbohydrate Motif	Glycosylated Ligand	Molecular Mechanism	Role in GB	Ref.
**Galectin-1**	Endothelial cells Astrocytes APCs Treg	Lactose poly-N-acetyllactosamine	*Immune markers* (CD3, CD4, CD7, CD43, CD45, CD69)	Activation of Fas-induced death, mitochondria apoptotic pathway, VEGF-R2/NRP-1; STAT/JAK1-2; c-Jun/AP-1; Lck/ZAP-70 Differentiation of IL-27/IL-10-producing tDCs Increased expression of IL-10 and IL-21 Modulation of the c-Maf/aryl receptor pathway Loss of mitochondrial membrane potential Release of cytocrome c	Tumor progression Angiogenesis Macrophage differentiation MDSC recruitment	[21,132,135,138,139,140,141,142,143,144,145,146]
**Galectin-3**	Endothelial cells Activated microglia Activated astrocytes Myeloid cells Fibroblasts	Lactose N-acetyllactosamine	*Cell adhesion/matrix* (Laminin, Vitronectin, Collagen I/IV, MCAM) *Immune markers* (TCR complex, CD7, CD29, CD45, CD71, LFA-1, TLR-4, LAG-3, CTLA-4) VEGF-R2	Inhibition of NKp30 signaling pathway Interferes with MICA-NKG2D affinity Increases/impairs cell-matrix adhesion (integrins) Activation of GSK-3β, RAS/PI3K/AKT, MEK/ERK Modulation of β-catenin and RAS/Bcl-2/Myc Increases Akt activity Stabilization of TGF- βR and signaling Supports IL-6 production	Proliferation Motility Resistance to radiotherapy	[21,130,133,134,135,137,156,157,158,160,161,162]
**Galectin-8**	Endothelial cells	α-(2-3)-Sialic acid Lactose N-acetyllactosamine	*Cell adhesion/ matrix* (Laminin, Fibronectin, Vitronectin, Collagen IV, CD166, Integrins α1β1/α3β1/ α5β1/α6β1) *Immune markers* (IL- 2R, TGF-βr, CD44)	Activation of VEGF-R2/NRP-1 and integrin-mediated signaling pathway Modulation of TGF- βR and IL-2R signaling pathway Promotes Treg differentiation and proliferation through STAT5 and Smad3 phosphorylation	Proliferation Prevent apoptosis Migration	[21,85,133,134,135,164,165,166,167,168,169]
**Galectin-9**	Activated astrocytes Microglia Endothelial cells	Lactose N-acetyllactosamine Forssman pentasaccharide	*Immune markers* (TIM3, DECTIN-1, CD44, VISTA, CD274, PD-L1, IDO1, LAG3) β3-interini Glut-2	TGF-β1- induced Treg differentiation Activation of VEGF-R2/NRP-1 pathway Promotes Smad3 phosphorylation Promote expansion of CD11b^+^Ly-6G^+^ MDSCs Upregulation of caspase-1, Granzyme B and Bid Downregulation of Lck and Bat3 signaling	Block T helper 17 Expansion of FoxP3^+^ Treg Apoptosis T cell exhaustion M2 polarization	[21,133,134,135,152,153,154,155]

## 5. Conclusions

Despite medical innovation, GB remains one of the most fatal malignant tumors [1].

Novel therapeutic strategies targeting GB cell self-renewal and growth as well as immunotherapies potentiating/eliciting anti-tumor immune response are currently investigated. However, unsatisfactory results have been reached with little impact on the availability of novel therapies [170,171].

Thus far, it has become clear that most therapeutic interventions aimed to dampen tumor growth, promote anti-tumor response as a bystander effect by reducing immune suppression and/or enhancing antigen presentation and eliciting T cell responses [172,173,174]. This indirect immune modulation contributes to the overall clinical benefit induced by the therapy [175].

It is also clear that the immunosuppressive, tumor-promoting microenvironment halts the drug-induced immunological benefits or the efficacy of immunotherapeutic interventions as such as Immune checkpoint blockade, now standard of therapy for several tumors, or more experimental immunotherapies as vaccination, adoptive T cell therapy, CAR-T [176,177,178].

In this scientific and biological framework, the tumor associated glycans that are generated by aberrant tumor glycosylation acquire a crucial importance for their immune-coding significance, and not just as markers of tumor phenotype and invasiveness [79,103].

Indeed, the lectin immune receptors that selectively sense and recognize these tumor associated structures have emerged as markers and possible therapeutic targets to dismantle the immunosuppressive networks [179].

The GB TME is characterized by a peculiar array of diverse myeloid cell types, whose molecular profile and functions are a challenging research field [180].

The diverse expression profiles of lectins displayed by these cell subsets is strongly suggestive of their role in the triggering and fueling of immunosuppression. Indeed, several compounds have been developed to block galectin- and Siglec-mediated interactions for therapeutic purposes in tumor and other pathological settings [181,182,183]. Glycomimetic peptides have been successfully employed for such purpose [182,184]; indeed the MGL-Tn glycan axis can be targeted with therapeutic benefit in ovarian cancer settings [116]. The availability of novel glycoproteomic approaches may allow in depth glycosylation profiling and mapping to identify the glycoconjugated ligands that in vivo participate to the lectin mediated immune networks in GB. Therefore, further investigations are required in order to better understand and clarify the contribution of these interactions in GB development and progression.

Glycan–lectin interactions mediate also immune cell migration [185]. Indeed, the matrisome has emerged as a key factor in the motility of tumor as well as immune cells and its modulation has been regarded as potential therapeutic option [101,186]. From an immunological point of view, the knowledge of the mechanisms that regulate immune cells infiltration is essential to switch the immune tumor contexture and improve the efficacy of treatments [187,188].

The characterization of the changes in glycan composition and the unveiling of the mechanisms of glycan-lectin interactions that occur in GB can critically contribute to identify novel therapeutic strategies and ameliorate already available immunotherapeutic treatment in GB.

## Figures and Tables

**Figure 1 ijms-23-06312-f001:**
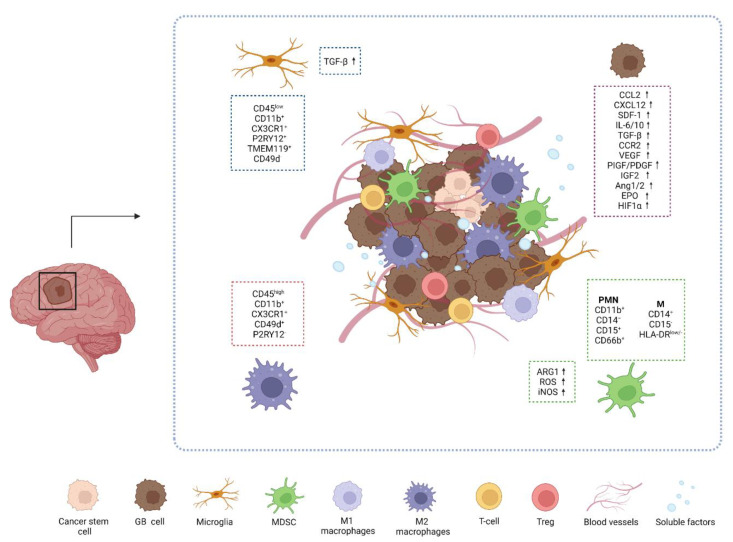
Schematic representation of glioblastoma immune microenvironment. Glioblastoma is highly vascularized with impaired circulation and hypoxic regions. Several soluble factors, as HIF-1α and VEGF, are involved in GB development and progression promoting EMT, ECM remodeling and immunosuppression through TAM and MDSC recruitment. Within tumor cells and cancer stem cells, myeloid cells are the most represented cell type in GB TME. TAM population includes macrophages (CD45^high^ CD11b^+^ CX3CR1^+^ CD49d^+^ P2RY12^−^) and MG (CD45^low^ CD11b^+^ CX3CR1^+^ P2RY12^+^ TMEM119^+^ CD49d^−^) while MDSC are mainly represented by PMN-MDSC (CD11b^+^ CD14^−^ CD15^+^ CD66b^+^). Created with BioRender.com, accessed on 2 June 2022.

**Figure 2 ijms-23-06312-f002:**
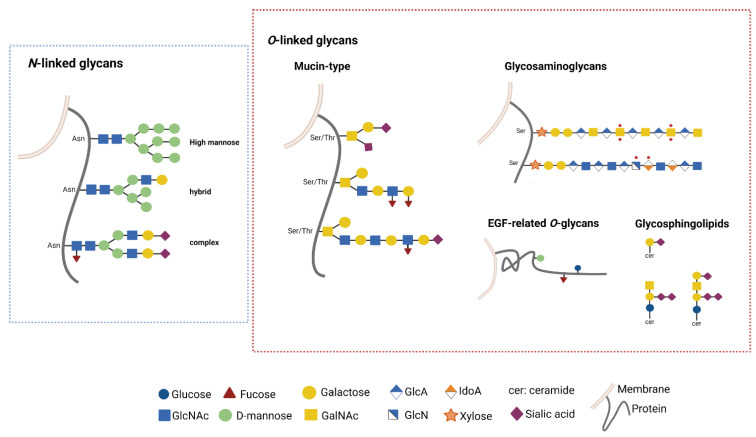
Common *N*- and *O*-glycan structures. Extracellular proteins are mainly glycosylated by the addition of monosaccharides in *N*- or *O*-linkage. *N*-glycans share a common core region in which the oligosaccharide *N*-Acetyl-D-glucosamine (GlcNAc) is covalently linked to the asparagine (Asn) residue. The elongation process through the linkage of D-mannose (Man) and the addition of other motifs, as such as Sia and Gal, led to the generation of High mannose, hybrid and complex *N*-glycoconjugates. *O*-glycosylation starts from the binding of monosaccharides, as such as *N*-acetyl-D-galactosamine (GalNAc) to serin/threonin (Ser/Thr) residues. GalNAc represents the substrate for further elongation of Mucin-type *O*-glycans. Other *O*-glycan structures include *O*-xylose (Xyl) for glycosaminoglycans, *O*- D-glucose (Glc) for glycosphingolipids, and *O*-Glc, fucose (Fuc) or Man for the EGF-related *O*-glycans. Created with BioRender.com, accessed on 2 June 2022.

**Figure 3 ijms-23-06312-f003:**
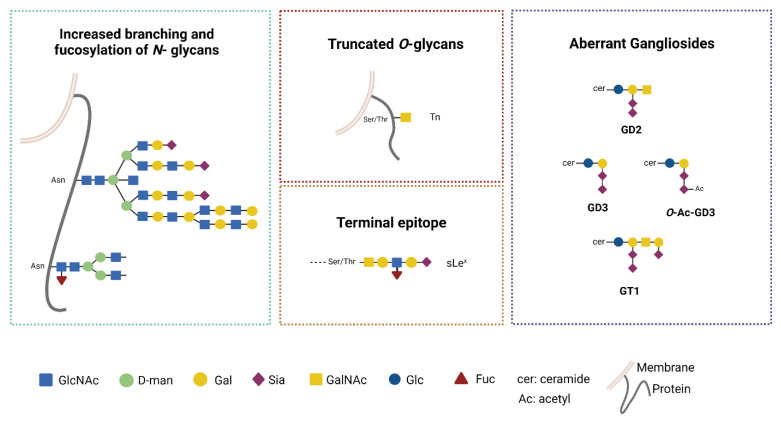
Common altered glycan structures in GB. Glycosylation changes are features of tumorigenesis and sustain tumor progression by affecting several physiological processes. Alterations in *N*-linked glycosylation, such as increased branching to form complex/hybrid structures and aberrant fucosylation, are mainly found in GB. Truncated *O*-glycans (Tn, sTn, T, ST) are typical of carcinomas and only Tn has been detected in GB. Excessive sialylation leads to the generation of Sialyl-Lewis Acid (S-Lewis X). Gangliosides are altered in GB and are enriched in GD2, GD3 and its acetylated form, GT1. Created with BioRender.com, accessed on 2 June 2022.

**Figure 4 ijms-23-06312-f004:**
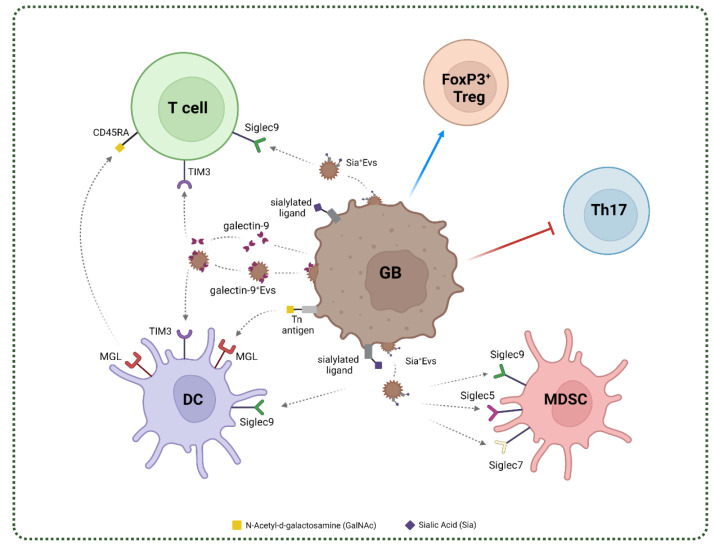
Schematic representation of lectin-mediated immunosuppressive networks in GB. Tumor cells modulate the microenvironment promoting immunosuppressive mechanisms that contribute to GB progression and evasion. Tumor cells promote FoxP3^+^Treg cells expansion and block T helper 17 polarization. Tumor cells express and release as soluble factors or through EVs galectin-9 that interacts with TIM3 receptor expressed on T cells and DCs, blocking their effector functions. Tumor cells show aberrant glycosylation of membran-bound proteins, thus generating truncated Tn-antigen and sialylated glycans. Tn-antigen directly interacts with MGL-expressing DCs. Moreover, DCs block T cells through MGL-CD45RA interaction. Sialylated glycans on tumor cells surface or EVs carrying sialylated proteins are recognized by Siglec5 and 7 on MDSCs and by Siglec 9 on MDSCs, DCs and T cells. Created with BioRender.com, accessed on 2 June 2022.

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
