# Peer review of "Glycan-Lectin Interactions as Novel Immunosuppression Drivers in Glioblastoma"

_ijms, 2022, doi:10.3390/ijms23116312_

Round 1

Reviewer 1 Report

Summary

The review provides an interesting aspect of immune evasion or suppression in cancers mediated through aberrant glycoforms expressed on GB and their interaction with cell-type specific lectins. The authors have tried to discuss the aetiology of GB with special reference to various lectins expressed on immune cells that engages aberrant glycoforms on GB to then affect immunosuppression. The review appropriately discusses the role of lectin-glycan interaction as drivers of immunosuppression modulation, which is currently a hot topic in both basic research and clinical research to identify and develop novel therapeutic strategies to overcome cancer recurrence. Hence, the authors have been successful in presenting this topic to a wide range of scientific readers with a clear propaganda. However, I find some parts of the review to be convoluted and hence would like to suggest corrections for the same before it to be published.     

MAJOR COMMENTS:

  1. The authors have tried to focus on aberrant glycan changes and their interactions with lectins thus, mediating immunosuppressive responses in GB pathophysiology. The authors have primarily laid their attention on N- and O-linked glycan aberrations, however, they must describe the Glycosphingolipids (GSL) synthesis and in detail their role in the GB pathology. Indeed they have neglected fucosyl-GM1. GSL form a crucial class of glycans which has been long involved in affecting GB aggressiveness through aberrant RTK functioning and modulating their immune responses.
  2. In section 3 “Aberrant glycosylation and lectin-mediated immunosuppressive networks in GB”, the authors have provided scattered information regarding the types of glycans and aberrant glycan changes affecting GB. It would be better to consider modifying this section in a way to give more clarity to the general readers on the glycans biosynthetic process and then discussing about the aberrant glycans involved in GB pathology.
  3. I found the section on lectins and their association with glycans in GB progression convoluted. Although, the authors have provided a tabular summary on different lectins and their corresponding ligands, it would be better to concise the text just to avoid repetition. Nonetheless, it would be more informative if the authors provide additional information’s and data available on certain lectins such as, Galectins and GSL in cancer immunosuppression or modulation.
  4. It will be good to have an additional section in the tables providing information on the underlying molecular mechanism of immune modulation upon lectin and glycan interaction within TME, if already known. The mechanisms discussed in the text does not shed much information to the readers, as in case of, Galectins in modulating T-cell response. It would be nice if the authors provide a detailed mechanism on them.
  5. Finally, the authors would like to consider and cite some works from Ludger Johannes group on Galectins and T-cell modulation in cancer.

MINOR CORRECTIONS:

  1. The author’s must recheck the text once again for grammatical errors and must consider another around of proof-reading.

Author Response

Comments and Suggestions for Authors. Summary

The review provides an interesting aspect of immune evasion or suppression in cancers mediated through aberrant glycoforms expressed on GB and their interaction with cell-type specific lectins. The authors have tried to discuss the aetiology of GB with special reference to various lectins expressed on immune cells that engages aberrant glycoforms on GB to then affect immunosuppression. The review appropriately discusses the role of lectin-glycan interaction as drivers of immunosuppression modulation, which is currently a hot topic in both basic research and clinical research to identify and develop novel therapeutic strategies to overcome cancer recurrence. Hence, the authors have been successful in presenting this topic to a wide range of scientific readers with a clear propaganda. However, I find some parts of the review to be convoluted and hence would like to suggest corrections for the same before it to be published.     

MAJOR COMMENTS:

  1. The authors have tried to focus on aberrant glycan changes and their interactions with lectins thus, mediating immunosuppressive responses in GB pathophysiology. The authors have primarily laid their attention on N- and O-linked glycan aberrations, however, they must describe the Glycosphingolipids (GSL) synthesis and in detail their role in the GB pathology. Indeed they have neglected fucosyl-GM1. GSL form a crucial class of glycans which has been long involved in affecting GB aggressiveness through aberrant RTK functioning and modulating their immune responses.

Response:

We thank the reviewer for this indication and agree with that. We therefore have included information regarding Glycosphingolipids and their role in GBM in Section 3 and integrating the references. Also, Figure 2 has been modified including both GalCer and GlcCer based GLS and a novel figure has been included (Figure 3) in which also the structure of GLS relevant in GB is outlined.

Action

Pag. 7, lines 239-247 (novel text and novel references)

Lipid glycosylation is  also crucial for the biology of cells. In particular,  glycosphingolipids (GLS), that are the major component of the outer cell plasmamembrane, are composed by a hydrophobic ceramide backbone linked to a first glycan moiety (Gal or Glc) through a glycosid linkage [67]. Complex enzyme network located among the intracellular compartments mediates the subsequent addition of other carbohydrate residues as such as GlcNAc, GalNAc, Fuc and Sia generating structurally different molecules which potentially correspond to a distinct function. Gangliosides (GG) are acidic GLS containing sialic acid residues, enriched in cell membrane microdomains, and are more abundantly found in the nervous system [68,69].

Pag. 10, lines 333-345 (novel text and novel references)

Also, glycolipid components are altered in GB. Early work had shown changes in GLS components in human gliomas, in particular ganglioside GD3 was shown to be increased in gliomas and correlated with malignancy [89,90]. So far, increase of GD2 and GD3 in GB has been associated to resistance to therapy and increase on invasiveness [91,92]. The aberrant GG pattern alters the organization of membrane microdomains with huge impact on the activity of tyrosin-kinase receptors and downstream cell signaling [93]. Indeed, GD3 plays a crucial role in the stemness and tumorigenicity of GB by activating c-MET signaling [94]. At the same time, GD3 has been shown to modulate innate immune response by specifically interacting with Siglec7 (see paragraph 4.2). More recently, increase of O-Ac-GD3 (GD3 carrying acetylated Sia) and the ganglioside GT1 were found in human GB samples, while GM1 was found in peritumoral brain tissue [95,96]. These differential patterns may be relevant to both tumor biology and immunomodulation mediated by GGs [97].

Pag. 6, lines 203-204

Figure 2 “Common N- and O-glycan structures” has been modified including the structure of GalCer based GLS.

Pag. 9, lines 294-301

Novel Figure 3 “Common altered glycan structures in GB” and its legend has been included.

  1. In section 3 “Aberrant glycosylation and lectin-mediated immunosuppressive networks in GB”, the authors have provided scattered information regarding the types of glycans and aberrant glycan changes affecting GB. It would be better to consider modifying this section in a way to give more clarity to the general readers on the glycans biosynthetic process and then discussing about the aberrant glycans involved in GB pathology.

Response:

We agree with the reviewer and thank for the request. The section 3 has been modified as requested. It has been organized in two paragraphs (3.1 and 3.2); text has been restructured and integrated with more information regarding general features of glycosylation pathways and GLS (section 3.1) and glycan alteration in GB (section 3.2) (see above).

Also, a novel figure showing the main glycosylation changes in GB has been included (now Figure 3)

Action

Pag.6, line 189

Title Section 3: 3.Aberrant glycosylation and lectin-mediated immunosuppressive networks in GB

Now it reads

  1. GB and glycosylation pathways

Pag. 6, line 190

Title paragraph 3.1 - 3.1 Glycosylation pathways

Pag. 8, line 261

Title paragraph 3.2 - 3.2 Aberrant glycosylation processes in GB

Pag. 6, lines 193-196 (novel text)

Indeed, specific enzymes partecipate to the finely tuned phases of the glycosylation process i.e. core extention, elongation and branching, and capping of the carbohydrate chains. This generates an astonishing diversity in the glycome that corresponds to a highly specific and selective biological functions [60,61].

Pag. 7, lines 221-225 (text moved from original 3.1 section)

In this pathway, the synthesis of GalNAc α1-O-Ser/Thr (the Tn antigen) is required for the next synthesis of the T antigen (Galβ1-3GalNAcα1-O-Ser/Thr) and further extension of long polylactosamine side chains that decor the protein backbone of mucins. These molecular structures function as protection and lubrification of epithelia and become truncated in carcinoma.

Pag. 7, lines 231-235

In original text Pag. 7, lines 219-221

 “Elongation processes give rise to a common pentasaccharide core region then further diversified in carbohydrate composition, on the basis of that, N-glycoconjugates are classified in High mannose, hybrid or complex [65,66]”

Now it reads:

Elongation processes give rise to a common pentasaccharide core region composed by GlcNAc and Man residues. This N-core structure is the substrate for the branching of the side chains that are further diversified in carbohydrate composition. On the basis of that, N-glycoconjugates are classified in High mannose, hybrid or complex [65,66].

Pag. 7-8, lines 248-259 (novel text)

Despite the diversity of glycosylation pathways, the carbohydrate chains can undergo to non-specific modifications thus generating molecular structures shared by distinct glycoconjugates. Sulfation, the addition of SO3 group to the carbohydrate backbone, is the most abundant glycan modification generating a large pattern of sulfated structures that play wide essential biological role. Moreover, the terminal addition of Sia (Sialylation) or Fuc (Fucosylation) are crucial events in the capping of the glycan chains, thus blocking the elongation process and generating glycan structures specifically recognized by glycan-binding proteins [60,70].

The variety and complexity of the glycome is shaped by the cellular stress and becomes a strategy for the cell to modulate its function and interactions with the microenvironment. Indeed, glycosylation alterations may be regarded as key features of diseases and a key factor for immune recognition [71].

Pag. 8, lines 261-368

              Novel section 3.2 (original 3.1) has been reorganized including the addition of novel Figure 3.

Now it reads as follow:

Aberrant changes in the glycosylation machinery early occur during tumor transformation i.e. increased N-glycan branching, truncated O-glycans and augmented O-glycan density on the protein backbone, and synthesis of neo-glycans carrying numerous terminal Sia or Fuc[19,72]. These aberrant glycosylation processes generate tumor associated glycans that play a key-role in tumorigenesis, metastasis and immune modulation [72,73].

So far, an accumulating body of evidences indicate that glycosylation changes occur in GB and involve a large variety of cellular substrates (proteins and lipids) and distinct metablic pathways thus contributing to the complex biology and impaired immunogenicity of GB (Figure 3) [18,74].

In GB, changes in N-glycosylation are characterized by altered branching, increased fucosylation and sialylation. Several receptors may be aberrantly N-glycosylated and this impacts on cell functions. N-acetylgalactosyltransferases (GnTs) are key enzymes for the N-glycan synthesis. MGAT1 (GnT1) and MGAT5 (GnT5) are up-regulated in glioma cells and in GB tissues [75,76]. MGAT1 plays a crucial role in the conversion of the carbohydrate core from High mannose to complex/hybrid structures.

This change increases the stability of the GLUT1 transporter which is N-glycosylated, promoting cell proliferation [75]. MGAT5 catalyzes the addition of terminal b-1,6-GlcNAc to the N-glycan and its overexpression in cancers supports N-glycan branching and correlates with tumor invasion [76,77]. Increased N-branching augments terminal sialylation and fucosylation processes that contributes to tumor spreading [78].

The tumor associated Sialyl-Lewis Acid (S-Lewis X) glycan generated by these changes is recognized by immune lectins [79]. Dysregulated fucosylation is a feature associated to GB aggressiveness. Fucosylation of the first GlcNAc residue of the N-core structure (Figure 3) is catalized by the fucosyltrasferases 8 (FUT8) (Figure 3). In human GB FUT8 overexpression isassociated with high tumor grade and aggressive disease. Interestingly, MET and EGFR molecules are substrate for FUT8 and the altered glycosylation increases their function with an overall gain of motility and proliferation for the tumor cells [80].

Indeed, inhibition of N-linked glycosylation has been proposed as an appealing pharmacological strategy to dampen receptor kinase activity and enhance radiosensitivity of GB cells [81].

The increased sialylation levels observed in glioma cells are mainly due to the overexpression of the a-2,3-sialyl transferase (ST3GAL1). Its up-regulation correlates with poor prognosis in GB patients and is associated to enhanced invasive potential [82]. On the contrary, a-2,6-sialylation is downregulated in the tumor cells , but it is upregulated in the GB vasculature, suggesting its crucial role for endothelial survival [83–85]. Therefore, two distinct sialylation pathways underlie distinct biological mechanisms for GB progression [74].Little is known about changes in the O-linked mucin type glycosylation. In carcinoma, alterations of this enzymatic pathway generate truncated O- glycans i.e. Tn and T and their sialylated version STn (NeuAcα2-6GalNAcα1-O-Ser/Thr) and ST (NeuAcα2-3Galβ1-3GalNAcα1-O-Ser/Thr) [86]. In GB, alteration of the enzymatic O-glycosylation pathway was found and Tn glycan expression was revealed by the binding of anti-Tn lectins in mouse model as well as in human tumor cells [87,88]. In GB mouse model the overexpression of Tn has been associated to strong immunosuppression (as described in section 4.1) [88].

Also glycolipid components are altered in GB. Early work had shown changes in GLS components in human gliomas, in particular ganglioside GD3 was shown to be increased in gliomas and correlated with malignancy [89,90]. So far, increase of GD2 and GD3 in GB has been associated to resistance to therapy and increase on invasiveness [91,92]. The aberrant GG pattern alters the organization of membrane microdomains with huge impact on the activity of tyrosin-kinase receptors and downstream cell signaling [93]. Indeed, GD3 plays a crucial role in the stemness and tumorigenicity of GB by activating c-MET signaling [94]. At the same time, GD3 has been shown to modulate innate immune response by specifically interacting with Siglec7 (see paragraph 4.2). More recently, increase of O-Ac-GD3 (GD3 carrying acetylated Sia) and the ganglioside GT1 were found in human GB samples, while GM1 was found in peritumoral brain tissue [95,96]. These differential patterns may be relevant to both tumor biology and immunomodulation mediated by GGs [97].

Another crucial source of glycan diversity is the matrisome. In GB, the ECM is increased due to the up-regulated secretion of PGs and the associated O-linked GAGs [98–100]. These ECM components can undergo to aberrant sulfation processes that appear to be relevant for the glycan alteration in GB ECM. Modulation of sulfation among distinct PGs was found and several core matrisome PGs (as such as decorin, agrin, glypcan-1), and laminin, tenascin, fibronectin among the others, were differentially regulated in tumor vs control tissues. Interestingly, changes in PG composition were also found within the GB samples in accordance to the IDH mutant status. From this analysis, O-glycosylated peptides from PGs were found only in the GB tissue and not in the control, again indicating that dysregulated glycosylation process generates specific glyco-profile [101].

Indeed, a glycoproteomic fingerprinting would be of great interest to in depth characterize the ECM components to unveil biological mechanisms aiding cell growth and metastatic spreading that could be useful as potential therapeutic targets [74,102].

  1. I found the section on lectins and their association with glycans in GB progression convoluted. Although, the authors have provided a tabular summary on different lectins and their corresponding ligands, it would be better to concise the text just to avoid repetition. Nonetheless, it would be more informative if the authors provide additional information’s and data available on certain lectins such as, Galectins and GSL in cancer immunosuppression or modulation.

Response:

We agree with the reviewer and concise the section 4, deleting the repetitions from the text when the information is reported in the Tables.

In particular, we have modified paragraph 4.3- Galectins in GB, deleting and rephrasing the original text and integrating it with description of the general role of galectins in cancer immunosuppression. Novel references have also been cited.

Action:

Original Manuscript Pag 13, lines 398-409

Interestingly, in U87MG glioma cells it has been shown that MAG is able to reduce the migratory capacity of these cells in culture [113]. The addition of sialic acids residues represents the last step of the glycosylation process resulting in a wide range of Sia-decorated glycoconjugates.Alterations in the sialylation pattern and sialyltransferases expression and activity are common in several disease processes, such as chronic inflammatory diseases and cancer [114]. As mentioned above, changes in glycosylation are considered as features of carcinogenesis [15]. Tumor cells can express aberrant amounts of sialoglycans on their surface which mediate the interaction with the TME and can function as an escape mechanism from the immune recognition [105]. It has been shown that in tumor models, sialoglycan-Siglecs interactions are able to impair and suppress the effector immune cells, such as NK and CD8+ T cells.

Pag. 15, lines 469-480

Interestingly, in U87MG glioma cells it has been shown that MAG is able to reduce the migratory capacity of these cells in culture [126].

It has been shown that in tumor models, sialoglycan-Siglecs interactions are able to impair and suppress the effector immune cells, such as NK and CD8+ T cells.

Pag. 16, lines 512-529

Original manuscript Pag. 14, line 440-444 has been integrated and rephrased

In TME, galectins have been shown to play a key role in tumor progression, promoting angiogenesis, ECM remodelling and immunosuppression, through the interaction with highly glycosylated immune-related ligands, such as CD4, CD7, CD43, CD45, TIM3. This engagement results in the induction of tolerogenic DC, CD4+ and CD8+T cell apoptosis and exhaustion, Treg cells expansion and NK impairment [21].

Now it reads

Interestingly, galectins exert their function both intracellularly and extracellularly [133]. Galectins have been shown to play a key role in tumor progression, promoting angiogenesis and ECM remodelling through the binding with highly glycosylated ligands expressed by several cell types. Tumoral galectins, interacting with glycosylated immune-related ligands, induce immunosuppression promoting tolerogenic DCs, CD4+ and CD8+T cell apoptosis and exhaustion, FoxP3+ Treg cells expansion and NK impairment [21]. Upon interaction with galectins, glycoproteins and glycolipids undergo to structural rearrangement, thus modulating their function and the downstream signalling pathways [133]. Galectin-1 shapes the immune compartment, promoting the proliferation of tolerogenic DCs and M2-macrophages, the apoptosis of Th1 and Th17 T cells and the expansion of Treg cells. Moreover, galectin-3 exerts its immunomodulatory functions affecting effector T cells activity by altering the immunological synapses organization and distancing TCR from CD8. These molecular events induce anergy, exhaustion and suppression of T cells. At the same time, galectin-3 promotes immunosuppressive cells, as Tregs and MDSCs, thus dampening the anti-tumor immune response [133–135].

Pag.17, lines 535-537

Indeed galectin-1, -3, -8 and -9 are expressed in GB and the interaction with their respective glycan-carrying molecules blocks the anti-tumor response, thus promoting immunosuppression (Table 2) [138]. 

Pag. 19, lines 577-584

Modification of the Original manuscript pag. 15-16, lines 485-491

The major mechanism through which galectin-9 exert its immunosuppressive mechanisms involves TIM3 binding. TIM3 is widely expressed on immune cells, especially on T-cells. The engagement with galectin-9 is known to block T helper 17 polarization, to stimulate the expansion of Fox-P3+ Treg and to induce apoptosis or exhaustion of pro-inflammatory T cells [21]. Other receptors that are recognized by galectin-9 are immune checkpoint molecules, as PD-L1, CTLA-4, IDO1 and LAG3, therefore galectin-9 is extremely relevant for immunosuppressive mechanisms [21].

Now it reads

The major mechanism through which galectin-9 exerts its immunosuppressive mechanisms involves the binding to TIM3, that is widely expressed on immune cells, especially on T-cells. [21]. Moreover, other immune checkpoints are recognized by galectin-9, therefore galectin-9 is extremely relevant for immunosuppressive mechanisms (see Table 2) [21].

Pag. 19-20, lines 590-607

Original manuscript pag 16, line 497-509 has been modified

Taken together, these studies suggest that galectin-9/TIM3 interaction can be one of the key mechanisms that sustains immunosuppression in GB TME and its selective targeting can have prognostic effects [138]. Despite the intrinsic function of this lectin in GB tumor cells has not been elucidated yet, its ability to affect the GB immune microenvironment offers galectin-9 as a potential immunotherapeutic target [139]. Notably, galectin-9 exerts its functions also when release in EVs. A fascinating study underlined the importance of GB derived-exosomes (GB-Exos) in immune escape. GB-Exos are enriched in unique proteins missing in other EVs and, among them, exosomal galectin-9 contributes to tumor progression by impairing DC and CD8+ T cells function through the interaction with TIM3 receptor. In vitro, lack of TIM3 in DC cocultured with GB cells resulted in an higher activation of DC, highlighting how the inhibition of DC by exosomal galectin-9 is TIM3 dependent and the loss of this inhibitory function can restore the anticancer immunity in GB patients [140].

Now it reads:

Notably, galectin-9 exerts its functions also when release in EVs. A fascinating study underlined that GB derived-exosomes (GB-Exos), enriched in galectin-9,  contribute to tumor progression by impairing DCs and CD8+ T cells function by TIM3 binding. Indeed, exosomal galectin-9 activity on DCs is TIM3 dependent and knockout of TIM3 in DCs, restore DC function and activation [152]. Taken together, these studies suggest that galectin-9/TIM3 can be one of the key mechanisms that sustains immunosuppression in GB TME and it has been proposed as therapeutical intervention for targeting cell-cell interaction and exosomes communication [153–155].

Pag. 20, lines 615-619

Modification of Original manuscript pag 16, lines 517-520

Galectin-3 has been also proposed as biomarker for differential grading and diagnosis of gliomas [131]. Recent studies have shown that galectin-3 promotes glioblastoma cells proliferation, motility and resistance to standard therapies and high levels are associated with a poor survival [132,133,148].

Now it reads:

Galectin-3 has been also proposed as biomarker for differential grading and diagnosis of gliomas [159], resistance to standard therapies and high levels are associated with a poor survival [160–162].

  1. It will be good to have an additional section in the tables providing information on the underlying molecular mechanism of immune modulation upon lectin and glycan interaction within TME, if already known. The mechanisms discussed in the text does not shed much information to the readers, as in case of, Galectins in modulating T-cell response. It would be nice if the authors provide a detailed mechanism on them.

Response:

We thank the reviewer for the suggestions. We integrated Table 1 and 2 with the information regarding known molecular mechanisms/signaling pathways involved in immune modulation for each lectin in an addition column indicated as “Molecular Mechanisms”.

Action:

Table Titles have been modified accordingly i.e.:

Pag. 13, line 448

Table 1. Classification, Expression, Binding Preference, Glycosylated Ligand, Known Molecular Mechanisms and role of C-type lectins and Siglecs in GB

Pag. 18, line 550

Table 2. Classification, Expression, Binding Preference, Glycosylated Ligand, Known Molecular Mechanisms and role of galectins in GB.

.

  1. Finally, the authors would like to consider and cite some works from Ludger Johannes group on Galectins and T-cell modulation in cancer.

Response:

We thank the reviewer for this suggestion

Action:

In the paragraph 4.3 and in the Table 2, we included the following references from Ludger Johannes group as requested:

[133] Johannes L, Jacob R, Leffler H. Galectins at a glance 2018:1–9. https://doi.org/10.1242/jcs.208884.

[134] Gilson RC, Gunasinghe SD, Johannes L, Gaus K. Galectin-3 modulation of T-cell activation: mechanisms of membrane remodelling. Prog Lipid Res 2019;76:101010. https://doi.org/10.1016/j.plipres.2019.101010.

MINOR CORRECTIONS:

  1. The author’s must recheck the text once again for grammatical errors and must consider another around of proof-reading.

Action:

English has been revised.

Reviewer 2 Report

In this article, the authors reviewed and summarized about the recent knowledge of cell surface glycans and lectins as the immunosuppression drivers in glioblastomas (GBs).

# Comments:

  • The titles of each chapter should be changed to more appropriate ones. For example, in the chapter “ Aberrant glycosylation and lectin-mediated immunosuppressive networks in GB”, the description about lectin-mediated immunosuppressive networks in GB is rarely found, and the title of next chapter “4. Lectin-mediated immunosuppressive networks in GB” is almost same as the title of previous chapter mentioned above. In addition, “Aberrant glycosylation” should be changed as “Aberrant protein glycosylation”. Therefore, in summary, the authors should correct the title “3. Aberrant glycosylation and lectin-mediated immunosuppressive networks in GB” into such as “3. Aberrant protein glycosylation in GB”.

  • The graphical summary of the general knowledge about each sequential glycosylation step of proteins should be added in the first half of manuscript.

  • The authors should explain the word “sialylation” and “fucosylation” in the first half of the chapter 3.

  • It is hard to understand where “glycan-lectin interactions in GB” is curated in the text. Therefore, the authors should modify the text to more understandable about this point.

Author Response

Comments and Suggestions for Authors

In this article, the authors reviewed and summarized about the recent knowledge of cell surface glycans and lectins as the immunosuppression drivers in glioblastomas (GBs).

# Comments:

  • The titles of each chapter should be changed to more appropriate ones. For example, in the chapter “Aberrant glycosylation and lectin-mediated immunosuppressive networks in GB”, the description about lectin-mediated immunosuppressive networks in GB is rarely found, and the title of next chapter “ Lectin-mediated immunosuppressive networks in GB” is almost same as the title of previous chapter mentioned above. In addition, “Aberrant glycosylation” should be changed as “Aberrant protein glycosylation”. Therefore, in summary, the authors should correct the title “3. Aberrant glycosylation and lectin-mediated immunosuppressive networks in GB” into such as “3. Aberrant protein glycosylation in GB”.

Response:

We thank the reviewer for this suggestion. Accordingly, we have modified the title of section 3 and organized it in two paragraphs

Action:

Pag.6, line 189

Title Section 3: 3.Aberrant glycosylation and lectin-mediated immunosuppressive networks in GB

Now it reads

GB and glycosylation pathways

Pag.6, line 190

Title paragraph 3.1 - 3.1 Glycosylation pathways

Pag.8, line 261

Title paragraph 3.2 - 3.2 Aberrant glycosylation processes in GB

  • The graphical summary of the general knowledge about each sequential glycosylation step of proteins should be added in the first half of manuscript.

Response:

We thank the reviewer for the comment and partially agree. We think that a detailed graphical summary for each different glycosylation pathway would add an unnecessary complexity to the manuscript.

However, we agree with the reviewer to improve section 3 for a better understanding of the basic glycobiology relevant to immune networks in GB.

We found more focused to our aim to depict the glycan structures that commonly occur in GB and are involved in Tumor-immune lectin interactions and we included novel Figure 3 on the aberrant glycan structures that commonly occur in GB.

We have also restructured the entire section 3 in two paragraphs dedicated to glycosylation in physiological conditions (novel 3.1 paragraph) and in GB (now 3.2 paragraph) as above described, modifying the text and integrating more information and references regarding general features of glycosylation pathways and glycosphingolipids (GLS) (section 3.1) and glycan alterations in GB (section 3.2).

Action:

Pag 9, line 294-301

We added a new figure (Figure 3) in paragraph 3.2 regarding the main alterations of glycans found in GB and the corresponding figure legend.

Pag. 7, lines 239-247 (novel text and novel references)

Lipid glycosylation is also crucial for the biology of cells. In particular,  glycosphingolipids (GLS), that are the major component of the outer cell plasmamembrane, are composed by a hydrophobic ceramide backbone linked to a first glycan moiety (Gal or Glc) through a glycosid linkage [67]. Complex enzyme network located among the intracellular compartments mediates the subsequent addition of other carbohydrate residues as such as GlcNAc, GalNAc, Fuc and Sia generating structurally different molecules which potentially correspond to a distinct function. Gangliosides (GG) are acidic GLS containing sialic acid residues, enriched in cell membrane microdomains, and are more abundantly found in the nervous system [68,69].

Pag. 8, lines 261-368

              Novel section 3.2 (original 3.1) has been reorganized including the addition of novel Figure 3.

Now it reads as follow:

Aberrant changes in the glycosylation machinery early occur during tumor transformation i.e. increased N-glycan branching, truncated O-glycans and augmented O-glycan density on the protein backbone, and synthesis of neo-glycans carrying numerous terminal Sia or Fuc[19,72]. These aberrant glycosylation processes generate tumor associated glycans that play a key-role in tumorigenesis, metastasis and immune modulation [72,73].

So far, an accumulating body of evidences indicate that glycosylation changes occur in GB and involve a large variety of cellular substrates (proteins and lipids) and distinct metablic pathways thus contributing to the complex biology and impaired immunogenicity of GB (Figure 3) [18,74].

In GB, changes in N-glycosylation are characterized by altered branching, increased fucosylation and sialylation. Several receptors may be aberrantly N-glycosylated and this impacts on cell functions. N-acetylgalactosyltransferases (GnTs) are key enzymes for the N-glycan synthesis. MGAT1 (GnT1) and MGAT5 (GnT5) are up-regulated in glioma cells and in GB tissues [75,76]. MGAT1 plays a crucial role in the conversion of the carbohydrate core from High mannose to complex/hybrid structures.

This change increases the stability of the GLUT1 transporter which is N-glycosylated, promoting cell proliferation [75]. MGAT5 catalyzes the addition of terminal b-1,6-GlcNAc to the N-glycan and its overexpression in cancers supports N-glycan branching and correlates with tumor invasion [76,77]. Increased N-branching augments terminal sialylation and fucosylation processes that contributes to tumor spreading [78].

The tumor associated Sialyl-Lewis Acid (S-Lewis X) glycan generated by these changes is recognized by immune lectins [79]. Dysregulated fucosylation is a feature associated to GB aggressiveness. Fucosylation of the first GlcNAc residue of the N-core structure (Figure 3) is catalized by the fucosyltrasferases 8 (FUT8) (Figure 3). In human GB FUT8 overexpression is associated with high tumor grade and aggressive disease. Interestingly, MET and EGFR molecules are substrate for FUT8 and the altered glycosylation increases their function with an overall gain of motility and proliferation for the tumor cells [80].

Indeed, inhibition of N-linked glycosylation has been proposed as an appealing pharmacological strategy to dampen receptor kinase activity and enhance radiosensitivity of GB cells [81].

The increased sialylation levels observed in glioma cells are mainly due to the overexpression of the a-2,3-sialyl transferase (ST3GAL1). Its up-regulation correlates with poor prognosis in GB patients and is associated to enhanced invasive potential [82]. On the contrary, a-2,6-sialylation is downregulated in the tumor cells , but it is upregulated in the GB vasculature, suggesting its crucial role for endothelial survival [83–85]. Therefore, two distinct sialylation pathways underlie distinct biological mechanisms for GB progression [74].Little is known about changes in the O-linked mucin type glycosylation. In carcinoma, alterations of this enzymatic pathway generate truncated O- glycans i.e. Tn and T and their sialylated version STn (NeuAcα2-6GalNAcα1-O-Ser/Thr) and ST (NeuAcα2-3Galβ1-3GalNAcα1-O-Ser/Thr) [86]. In GB, alteration of the enzymatic O-glycosylation pathway was found and Tn glycan expression was revealed by the binding of anti-Tn lectins in mouse model as well as in human tumor cells [87,88]. In GB mouse model the overexpression of Tn has been associated to strong immunosuppression (as described in section 4.1) [88].

Also, glycolipid components are altered in GB. Early work had shown changes in GLS components in human gliomas, in particular ganglioside GD3 was shown to be increased in gliomas and correlated with malignancy [89,90]. So far, increase of GD2 and GD3 in GB has been associated to resistance to therapy and increase on invasiveness [91,92]. The aberrant GG pattern alters the organization of membrane microdomains with huge impact on the activity of tyrosin-kinase receptors and downstream cell signaling [93]. Indeed, GD3 plays a crucial role in the stemness and tumorigenicity of GB by activating c-MET signaling [94]. At the same time, GD3 has been shown to modulate innate immune response by specifically interacting with Siglec7 (see paragraph 4.2). More recently, increase of O-Ac-GD3 (GD3 carrying acetylated Sia) and the ganglioside GT1 were found in human GB samples, while GM1 was found in peritumoral brain tissue [95,96]. These differential patterns may be relevant to both tumor biology and immunomodulation mediated by GGs [97].

Another crucial source of glycan diversity is the matrisome. In GB, the ECM is increased due to the up-regulated secretion of PGs and the associated O-linked GAGs [98–100]. These ECM components can undergo to aberrant sulfation processes that appear to be relevant for the glycan alteration in GB ECM. Modulation of sulfation among distinct PGs was found and several core matrisome PGs (as such as decorin, agrin, glypcan-1), and laminin, tenascin, fibronectin among the others, were differentially regulated in tumor vs control tissues. Interestingly, changes in PG composition were also found within the GB samples in accordance to the IDH mutant status. From this analysis, O-glycosylated peptides from PGs were found only in the GB tissue and not in the control, again indicating that dysregulated glycosylation process generates specific glyco-profile [101].

Indeed, a glycoproteomic fingerprinting would be of great interest to in depth characterize the ECM components to unveil biological mechanisms aiding cell growth and metastatic spreading that could be useful as potential therapeutic targets [74,102].

  • The authors should explain the word “sialylation” and “fucosylation” in the first half of the chapter 3.

Response:

We agree with the reviewer request.

Action:

The following sentences have been added:

Pag. 7, lines 248-259

Despite the diversity of glycosylation pathways, the carbohydrate chains can undergo to non-specific modifications thus generating molecular structures shared by distinct glycoconjugates. Sulfation, the addition of SO3 group to the carbohydrate backbone, is the most abundant glycan modification generating a large pattern of sulfated structures that play wide essential biological role. Moreover, the terminal addition of Sia (Sialylation) or Fuc (Fucosylation) are crucial events in the capping of the glycan chains, thus blocking the elongation process and generating glycan structures specifically recognized by glycan-binding proteins [60,70].

The variety and complexity of the glycome is shaped by the cellular stress and becomes a strategy for the cell to modulate its function and interactions with the microenvironment. Indeed, glycosylation alterations may be regarded as key features of diseases and a key factor for immune recognition [71].

  • It is hard to understand where “glycan-lectin interactions in GB” is curated in the text. Therefore, the authors should modify the text to more understandable about this point.

Response:

We thank the reviewer for this comment. We have modified and rephrased throughout the manuscript to better clarify this point and provide more understandable information to the reader. In particular, specific modifications are reported below

Action:

Pag. 11, line 372

The original title of section 4, 4- Lectin mediated immunosuppressive networks in GB has been modified.

Now it reads:

4- Glycan-Lectin interactions and immunosuppressive networks in GB.

Pag. 13, line 448 and Pag. 18, line 550

We modified the column “Ligand-carrying motif” of Tables 1 and 2 into “Glycosylated Ligand”.

Now it reads:

Table 1. Classification, Expression, Binding Preference, Glycosylated Ligand, Known Molecular Mechanisms and role of C-type lectins and Siglecs in GB

Table 2. Classification, Expression, Binding Preference, Glycosylated Ligand, Known Molecular Mechanisms and role of galectins in GB.

Pag. 16, lines 512-529

Interestingly, galectins exert their function both intracellularly and extracellularly [133]. Galectins have been shown to play a key role in tumor progression, promoting angiogenesis and ECM remodelling through the binding with highly glycosylated ligands expressed by several cell types. Tumoral galectins, interacting with glycosylated immune-related ligands, induce immunosuppression promoting tolerogenic DCs, CD4+ and CD8+T cell apoptosis and exhaustion, FoxP3+ Treg cells expansion and NK impairment [21]. Upon interaction with galectins, glycoproteins and glycolipids undergo to structural rearrangement, thus modulating their function and the downstream signalling pathways [133]. Galectin-1 shapes the immune compartment, promoting the proliferation of tolerogenic DCs and M2-macrophages, the apoptosis of Th1 and Th17 T cells and the expansion of Treg cells. Moreover, galectin-3 exerts its immunomodulatory functions affecting effector T cells activity by altering the immunological synapses organization and distancing TCR from CD8. These molecular events induce anergy, exhaustion and suppression of T cells. At the same time, galectin-3 promotes immunosuppressive cells, as Tregs and MDSCs, thus dampening the anti-tumor immune response [133–135].

Pag.17, lines 535-537

Indeed galectin-1, -3, -8 and -9 are expressed in GB and the interaction with their respective glycan-carrying molecules blocks the anti-tumor response, thus promoting immunosuppression (Table 2) [138]. 

Round 2

Reviewer 2 Report

I consider the authors responded to all my questions and requests properly.